# Design and Implementation of a Personalizable Alternative Mouse and Keyboard Interface for Individuals with Limited Upper Limb Mobility

Daniel Andreas *⬤, Hannah Six ⬤, Adna Bliek ⬤ and Philipp Beckerle ⬤

Chair of Autonomous Systems and Mechatronics, Friedrich-Alexander-Universität Erlangen-Nürnberg, 91052 Erlangen, Germany
* Correspondence: daniel.andreas@fau.de

**Abstract:** People with neuromuscular diseases often experience limited upper limb mobility, which makes the handling of standard computer mice and keyboards difficult. Due to the importance of computers in private and professional life, this work aims at implementing an alternative mouse and keyboard interface that will allow for their efficient use by people with a neuromuscular disease. Due to the strongly differing symptoms of these diseases, personalization on the hardware and software levels is the focus of our work. The presented mouse alternative is based on a spectacle frame with an integrated motion sensor for head tracking, which enables the control of the mouse cursor position; the keyboard alternative consists of ten keys, which are used to generate word suggestions for the user input. The interface was tested in a user study involving three participants without disabilities, which showed the general functionality of the system and potential room for improvement. With an average throughput of 1.56 bits per second achieved by the alternative mouse and typing speeds of 8.44 words per minute obtained using the alternative keyboard, the proposed interface could be a promising input device for people with limited upper limb mobility.

**Keywords:** personalization; assistive technology; alternative input device; keyboard replacement; hands-free mouse; vibrotactile feedback; neuromuscular disease

## 1. Introduction

Computers have become an essential part of many people's lives, especially in the past years when we increasingly relied on computers for communication during the pandemic. The ability to control the computer efficiently is important in the performance of basic tasks such as web browsing, text editing, or writing e-mails and is nowadays a necessity for many jobs. People with a neuromuscular disease often experience limited upper limb mobility, which can make operating the computer with a regular mouse and keyboard challenging. Friedreich's Ataxia alone affects approximately 1 out of 20,000 individuals in south-western Europe [1], not counting other neuromuscular diseases such as muscular dystrophy [2] or spinal muscular atrophy [3]. People with muscular dystrophy experience symptoms such as muscle weakness and loss of coordination, which can vastly vary between individuals, depending on genetics, the muscles being affected by the disease, and also the psychological motivation of the person [2]. Spinal muscular atrophy has an early onset age of six to eighteen months, which severely impacts the symptomatic course in individuals, ranging from the ability to walk during adulthood to the inability to sit [3]. People with a neuromuscular disease often suffer from dysarthria [4], which hinders them from using speech recognition to control the computer. Hence, this work focuses on the design and implementation of an alternative mouse and keyboard interface that can be controlled with limited upper limb mobility. Special emphasis is placed on the personalization of the alternative interface to account for the various symptoms experienced by the target population.

Tongue-operated devices provide reliable input and have previously been investigated by Manal et al. [5] and Yousefi et al. [6]. However, this input technique was disregarded in our work since it compromises the ability to speak during operation. Head motion tracking, on the other hand, is not impacted by that restriction and has already been successfully implemented by Fall et al. [7], who included an IMU into a headset. Within the case study of Abrams et al. [8], users of the target population were opposed to wearing a headband for head tracking. This might also apply to the headset used in the study of Fall et al. [7] due to the its unusual appearance. In contrast, wearing glasses is more socially accepted and is inobtrusive. Gür et al. [9] implemented an alternative mouse based on 3D printed spectacles, including an IMU for head motion tracking and force-sensitive resistors (FSR) that trigger mouse clicks by the activation of the Temporalis muscle. Chung et al. [10,11] took a similar approach to identifying food intake by integrating force sensors into the hinges of the spectacles in order to benefit from the leveraging effect of muscle movement on the temples while chewing. Similarly, Shin et al. [12] used a piezoelectric sensor mounted on the temple of the spectacle to sense food intake.

Gür et al. [9] combined the approach of using spectacles as the mouse alternative with an ambiguous keyboard, similar to the scanning ambiguous keyboard by MacKenzie and Felzer [13] or DualScribe by Felzer et al. [14]. Ambiguous keyboards have multiple letters assigned to each of the keys; a list of word suggestions from which the desired word can be selected is then compiled based on the input generated by the user. Gür et al. [9] used this approach in combination with force-sensitive resistors as adaptable keys, which have previously been shown to generate valid input signals in a study with one participant from the target population [8].

Due to its simplistic and promising design, we decided to improve the approach using the alternative mouse and keyboard interface by Gür et al. [9]. However, we have placed a strong emphasis on the personalization of the interface with respect to the software and hardware in order to make the system accessible to the largest audience possible, independent of each individual's anatomy and physiology. Another focus of our work is the implementation of a desktop application for the alternative mouse and keyboard interface in order to allow for their usage outside of confined test environments and to make them usable in daily tasks such as web browsing and text editing.

## 2. Concept and Implementation of a Personalizable Interface

The proposed mouse interface of this work makes use of head tracking to directly control the cursor position on the screen. The keyboard alternative is based on a ten-button layout to minimize finger movement during typing and relies on word suggestions. The novelty of this work lies in the high possibility for the personalization of the combined alternative mouse and keyboard interface, which has been designed for people with limited upper limb mobility. The interface can be adapted to match the individual preferences of the users. Adaptation to individual preferences is expected to highly influence user experience and usability. Therefore, the previous system of Gür et al. [9] was enhanced in terms of both hardware and software. The functionality of the individual components will be further described within the following subsections.

The concept of the alternative mouse and keyboard interface is described in Figure 1. The user is outfitted with spectacles that include sensors for controlling the cursor position on the screen and for activating mouse clicks. The generated signals are acquired by a microcontroller, which in this case is an Arduino Mega with an ATmega2560 processing unit that allows for real-time processing in this application. The microcontroller is connected directly to the computer via a USB cable. The alternative keyboard, consisting of ten buttons based on FSRs, is also directly wired to the microcontroller for signal processing. Vibration motors are placed where the wrist or forearms usually rest while typing, as shown in Figure 1, and these are used to provide haptic feedback to the user for each press of the button.

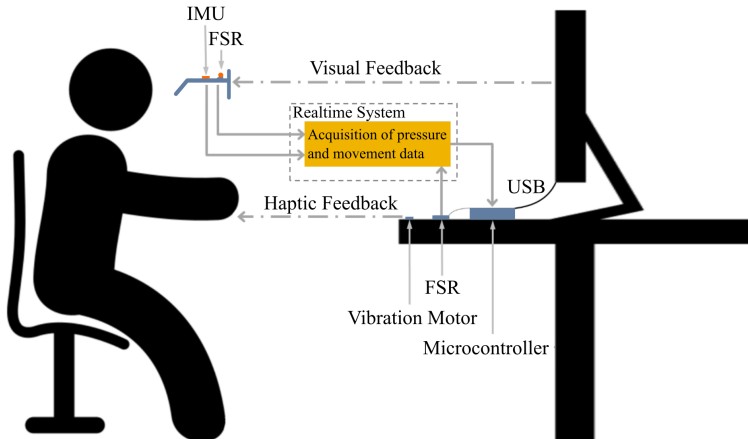

**Figure 1.** Concept of the alternative mouse and keyboard interface, showing the connection between the components and the user. The general setup is illustrated as intended during usage. The components of the alternative interface are marked in blue.

An image of the implemented alternative mouse and keyboard interface can be seen in Figure 2, which shows the actual arrangement of the components presented in Figure 1.

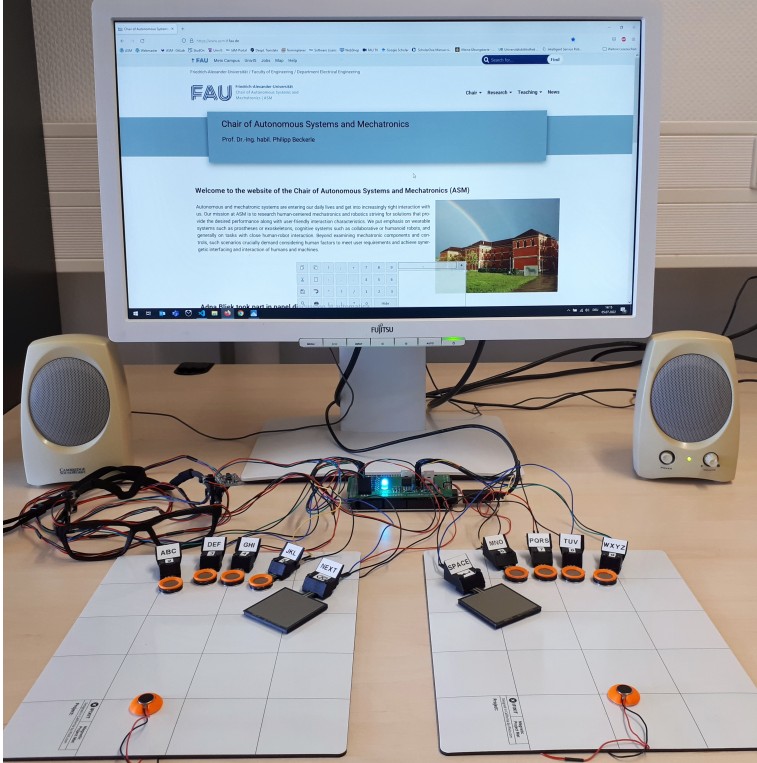

**Figure 2.** Image of the implemented alternative mouse and keyboard interface. The desktop application installed to be able to use the system for daily life tasks is active and creates a user interface in the lower center of the screen, with an opacity of 80%. The speakers on each side of the monitor are used to provide auditory feedback during the experiment and to introduce the user to each task through a brief instructional video. The 24″ monitor has a screen resolution of 1920 × 1080 pixels.

### 2.1. Mouse Alternative

The alternative mouse has been redesigned with respect to the frame for enhanced head tracking and most importantly, for greater adjustability of the overall fit in order to provide more reliable mouse click activation across users. The adjustments will be described in more detail in the mentioned order within this section. The frame of the

spectacles is now available in three different sizes, with the smallest frame measuring 130 mm in width (measured between both spectacle hinge screws), while the medium and large sizes being 5 % and 10 % wider, respectively. To allow for a less obstructive use of the mouse alternative, lighter cables are used to connect the sensors of the spectacles to the microcontroller, which significantly decreases the strain. Furthermore, all cables are guided to the microcontroller via the left side of the frame; therefore, all cables now come from one side, as shown in Figure 3, which facilitates taking the spectacles on and off.

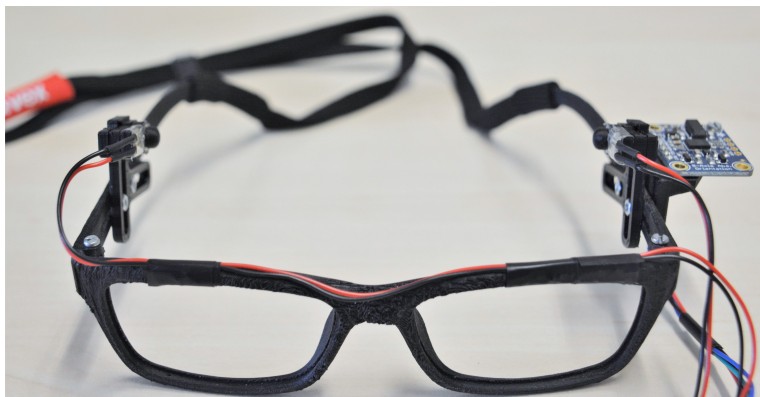

**Figure 3.** Image of the mouse alternative. The inertial measurement unit that tracks head motions for mouse cursor control is shown on the right. Force-sensing resistors are mounted onto each temple and can be adjusted in position to ensure good contact with the Temporalis muscle for mouse click activation. The spectacle strap is used to avoid involuntary movement of the spectacles caused by head motions or the activation of the Temporalis muscle.

An inertial measurement unit (IMU) is mounted onto the left temple of the spectacles, as shown in Figure 3. As suggested by Gür et al. [9], yaw and pitch head movements can thus be used to control the position of the mouse pointer on the screen. In addition to that, the option to scroll by rolling the head, as shown in Figure 4, has been added as a feature to improve user experience while reading text documents.

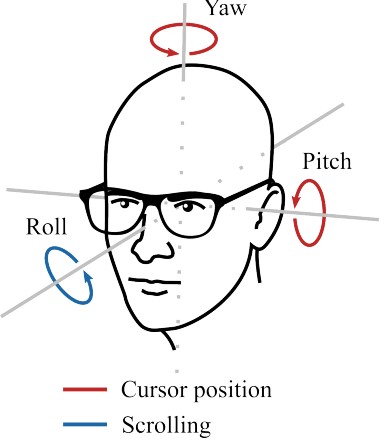

**Figure 4.** Yaw and pitch (red) motions determine mouse cursor position, while roll movements (blue) control up and down scrolling. Modified from Gür et al. [9].

Gür et al. [9] observed sensor drift in the MPU-6050 IMU by InvenSense Inc. due to involuntary human head movements during calibration. However, our tests with this IMU showed sensor drift over long periods regardless of the sensor being completely stationary during calibration and data acquisition. The BNO055 IMU by Bosch Sensortec GmbH proved to be less prone to sensor drift during the first tests after calibration and reached the best dynamic orientation accuracy of three consumer-grade IMUs in a test by

Lin et al. [15]. For our application, the calibration data are stored in the EEPROM of the microcontroller and are restored during the system boot-up. While there was no sensor drift during stationary use, sensor drift could still be experienced in motion over long periods. To account for this issue, we added a feature that allows the user to recenter the mouse cursor by activating the mouse for three seconds in a fixed position. Additionally, sudden cursor jumps were observed during testing, which were likely caused by disturbances in the magnetic field from surrounding devices and objects as well as the wiring on the spectacles. This issue could be resolved by disregarding magnetometer data from the internal sensor fusion and performing the calibration of the IMU while mounted on the spectacle frame.

Mouse clicks can be triggered through the activation of the masticatory muscles. Therefore, the two FSRs were placed on the Temporalis muscle, in the temporal region where the elevation due to muscle tension is most pronounced. This area varies between each person, and so a 3 *DoF* calibration plate, shown in Figure 5, was designed to place the FSRs at the optimal position to acquire the best possible signal. The calibration plate can be moved along the temple within a 13 mm range, through the recesses in the temple and the plate itself; it can also be tilted up or down within a range of 14 mm by two M2 screws. Furthermore, the firmness of the contact between the FSR and the skin over the Temporalis muscle can be adjusted through spacers that range between 1 mm and 5 mm, with 0.5 mm increments. This same extent was not possible with the previous design by Gür et al. [9]. The right image in Figure 5 shows an exploded view of how the calibration plate and the FSR are connected to each other, with the calibration plate at the bottom, the spacer in the middle, and the FSR support structure at the top. The support structure for the FSR contains a recess for an M2 nut and is secured by some tape that also fixates the FSR. A screw can be used to connect the three parts by putting it through the inserted nut. The spacer system was designed in a way that each component could be joined securely to the connecting parts without allowing for any unwanted movements during use and to assure reliable signals. This is still the case when no spacer is used and the FSR support is directly connected to the calibration plate.

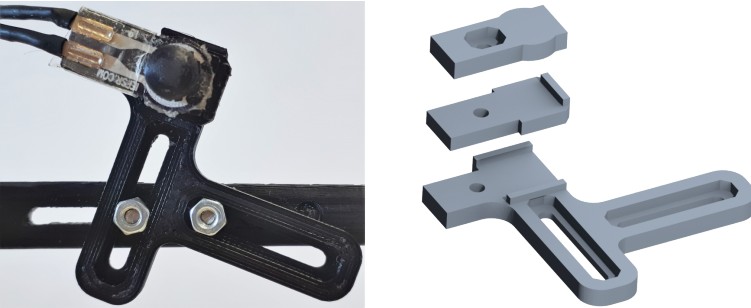

**Figure 5.** Image on the left shows the calibration system of the right temple in order to position the FSRs on the Temporalis muscle for mouse click activation. A dome-shaped plastic part is attached to the sensor to ensure good contact with the muscle. The right image shows an exploded view of the individual components to achieve the necessary adaptability.

For mouse click activation, the previously used Interlink 402 FSR, with an active sensor area of 14.7 mm, was replaced by the smaller Interlink 400 FSR in the short version, with an active area of only 5.6 mm, in order to decrease the weight and bulk of the spectacles. A dome-shaped structure was attached to the 5.6 mm active area of the FSR with the use of a double-sided adhesive tape to ensure good contact between the skin over the Temporalis muscle and the sensor, as well as to obtain the best possible signal quality and high sensitivity (see Figure 5 left). Lastly, a spectacle strap was added to the frame to avoid any unwanted movement of the spectacles that could occur during the activation of the masticatory muscles for mouse click activation (see Figure 3).

## 2.2. Keyboard Alternative

The ambiguous keyboard layout used in this work was partially adopted from works by Abrams et al. [8] and Gür et al. [9] and was based on the scanning ambiguous keyboard approach using a ten-key layout by MacKenzie et al. [13]. The latter eliminates the need for the user to move any of their fingers while typing as it takes into account the possible movement restrictions of the target group. Multiple letters are assigned to each key, as shown in Figure 6. To type "hello world", for instance, the user needs to enter "43556" first, which corresponds to the letter sequence of the word "hello". Since several words match this key input, the user then needs to select the desired word from a list of suggested words by pressing the next key with the left thumb. The word can then be entered by pressing the space button with the right thumb, which will also automatically enter a blank space behind the word so that the user could directly continue typing the next word "world" by putting in the sequence 96753.

Another added feature gives the user the ability for case-sensitive writing, which was not possible with the previous implementation by Gür et al. [9]. Capital letters can now be accessed by pressing long on the button of the respective key. Other functionalities that can now be accessed through a long button press are the clear button with the left thumb, which deletes the last character, and the enter button with the right thumb, which adds a blank line.

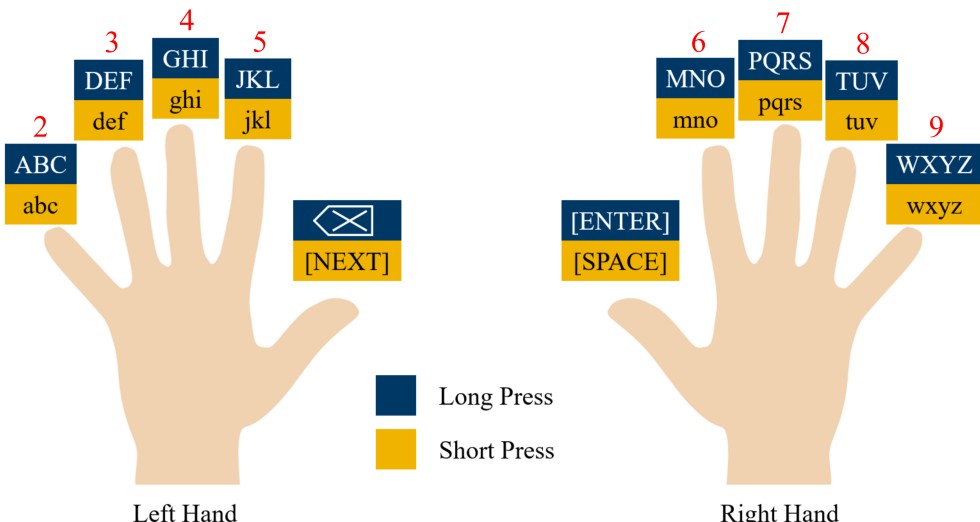

**Figure 6.** Keyboard layout showing the letter and function distribution for each finger. Secondary functionalities are in the blue boxes and can be accessed by long button presses. The red digits represent the underlying code for each button that are used to generate word suggestions.

A few modifications were made to the mechanical button design in order to improve the typing experience. Due to the flat shape of the FSRs, fingers could easily slip off the sensor. To resolve this issue, a support ring was designed and attached with an adhesive to the top of the Interlink 402 FSR without covering the 14.7 mm diameter active sensor area in the center, as shown in Figure 7, thus maintaining the sensor's sensitivity. Additionally, naming indicators were added to each of the buttons so that the user would know which letter or function keys were bound to which button at all times. Since hand and finger positions can highly vary between each user, the naming indicators were elevated and shifted backward in order not to be covered by the fingers while typing, which can be seen in Figure 7. The thumbs are positioned on the larger Interlink 406 FSRs, which have a square active area of 31.8 mm × 31.8 mm. Thus, a support structure to keep the fingers in place is not necessary here. To accommodate the varying hand shapes and sizes of different individuals, the buttons were then mounted onto a magnetic pad that allows the user to freely but stably arrange the buttons in order to adopt a comfortable hand position for longer use with less fatigue.

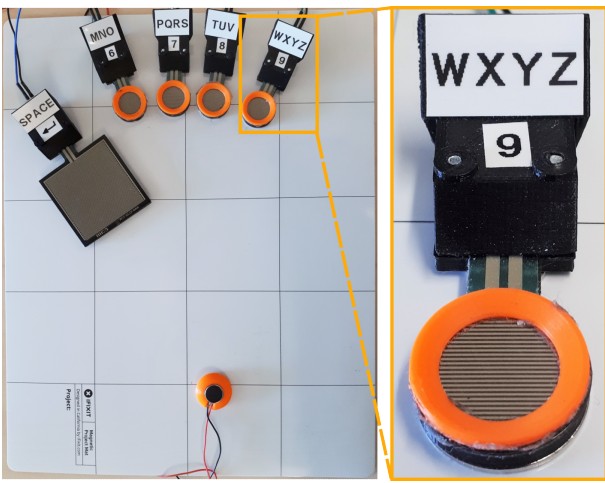

**Figure 7.** Magnetic pad and the buttons for the right hand as well as coin-shaped, eccentric, rotating mass vibration motors (MF-6318927) for haptic feedback, with a diameter of 10 mm and a height of 2.7 mm. On the right, a detailed view of a single button with a naming indicator and a support ring to prevent the finger from slipping off the button.

The libraries used to generate the dictionaries containing the keyboard input code, which is based on the digits for each word, are text files (*.txt) that contain a significant amount of words in the respective language. The file is formatted in such a way that each line contains a single word. Besides the keyboard input code, the frequency of each word being used is now also stored within the generated dictionary and is constantly updated in real time. Through that new feature, the dictionary adapts to the user while favoring more frequently used words in the word suggestion list, which is compiled based on the keyboard input code.

Users now have the option to save their modified libraries and load them directly using the setup window of the desktop application, which will be further described in the following subsections. This will also speed up the start-up phase of the application since the dictionary does not need to be generated for each new session.

Multiple-language settings are currently available with English [16] and German [17]. However, further languages could be added in the future, with little changes to the Python source code of the desktop application. To add a new language, the letter distribution similar to Figure 6 must first be defined, and a text file containing the majority of the words of the respective language has to be provided.

Table **??** contains the delays in compiling the list of word suggestions based on the keyboard input, measured with a Ryzen 5 4500U processor and 16 GB of RAM. The mean and standard deviation of the delays of all buttons representing different letters were computed for a single stroke of each button, as well as those for two strokes and three strokes of the same button; this was carried out for both the English and German dictionaries. As expected, the time required to compile word suggestions for the first keyboard input was the highest since it contained the largest amount of matching words, which continuously decreased for any additional keyboard input. The measured delay is also directly related to the size of the dictionary. In our tests, the German dictionary contained 2,152,669 words and the English one only 84,095, which explains the much higher computing time for the German dictionary. A study by Deber et al. [18] found that latency for tapping tasks through indirect input devices (no direct touch) becomes perceivable above 96 ms . This was also observed in the first tests using the alternative keyboard, where word suggestions appearing with a delay of roughly below 100 ms were almost imperceptible and thus not did have a significant influence on typing experience. Meanwhile, the delay in compiling the list of word suggestions for the first keyboard input using the German library was very noticeable and distracting at $526.14 \pm 125.51$ ms. The relatively high standard deviation

could possibly be explained by the impact of background tasks on the processing unit and might also impact the user experience while typing with the alternative keyboard.

**Table 1.** Mean and standard deviation of computing times in milliseconds, employed to compile the list of word suggestions that favor frequently used words for the first, second, and third keyboard inputs, using the German and English dictionaries in the standard and adjusted configurations.

| | English (ms) | | German (ms) | |
|---|---|---|---|---|
| | Standard | Adjusted | Standard | Adjusted |
| 1st keyboard input | $31.63 \pm 41.01$ | $0.00 \pm 0.00$ | $526.14 \pm 125.51$ | $0.00 \pm 0.00$ |
| 2nd keyboard input | $2.63 \pm 2.06$ | $2.85 \pm 2.54$ | $70.03 \pm 44.61$ | $111.67 \pm 78.07$ |
| 3rd keyboard input | $0.25 \pm 0.44$ | $0.25 \pm 0.44$ | $5.26 \pm 5.64$ | $6.50 \pm 6.53$ |

Since word suggestions for the first keyboard input are rarely helpful and require a lot of computational resources at the same time, the application was changed by adjusting the configuration to only suggest single characters that match the first keyboard input. Pressing button 6 from Figure 6 would thus suggest the characters m, n, and o. Word suggestions would only be compiled upon the second and all following keyboard inputs for each word. This eliminates the delays for the first keyboard inputs from Table **??** and merely impacts the already low and almost imperceptible delays for the second and third keyboard inputs of the standard configuration, making the alternative interface more accessible across platforms with less powerful processing units.

*2.3. Software Calibration*

Software calibration for each individual user is crucial for the system to work reliably and is especially important for people with a neuromuscular disease due to the variations in symptomatology. Therefore, we implemented a guided calibration process to find the optimal settings for each user, as shown in Figure 8, which will be further explained within this subsection.

The first calibration step includes the measurement of the initial force between the FSRs on the spectacle frame and the masticatory muscles. The force needs to be high enough to ensure good signal quality while still being comfortable over long periods of usage. Therefore, the force of each FSR is visualized as in Figure 8a. To assure optimal fit, the signal needs to be within the green area during a relaxed state of the masticatory muscle and within the red area during activation. Finally, the force during the relaxed state will be saved and used as a new baseline for all measured signals.

Next, the threshold at which a mouse click shall be triggered is calibrated. The user is asked to activate the masticatory muscle 10 times in a comfortable manner. The resulting signals are then used to determine the trigger and release threshold of a mouse click by multiplying the mean value of the peaks shown in Figure 8b by 0.7 for the trigger threshold, and then multiplying the resulting value by 0.5 for the release threshold of the mouse click in the default configuration. It should be noted that within the current setup, mouse clicks are triggered upon release.

To address possible variations in screen size, resolution, and distance between the user and the screen, the mouse cursor sensitivity needs to be calibrated accordingly. During the calibration procedure outlined in Figure 6b, the user is asked to first move their head in the direction of the lower left corner of the screen, followed by the activation of a mouse click using the alternative mouse, and repeating the same when moving the head to the upper right corner of the screen. The determined boundary values are then used to compute the cursor position $c$ on the screen based on the user's head movements with the following equation:

$$c = \frac{d}{|m - n|}(s \cdot u - n) + r_i \tag{1}$$

Here, $m$ and $n$ are vectors containing the maximum and minimum angles within the screen. Vector $d$ contains the horizontal and vertical display resolution in pixels, vector

$s$ has the sensor values of the IMU gyroscope that describe the head orientation in Euler angles, and $u$ is derived from the sensitivity than can be established during setup, resulting in a value between 0 and 2 when setting the cursor sensitivity to 0 % and 100 %, respectively. Lastly, $r_i$ denotes the correction vector for recalibrating the mouse cursor while being used. The values for recalibration $r_i$ are updated each time the user triggers a recalibration through a 3-s-long static mouse press using the spectacles and are updated according to the following equation:

$$r_i = \frac{d}{2} + p + r_{i-1} \tag{2}$$

with $r_{i-1}$ being the previous correction values for recalibration and $p$ being the current position of the cursor in pixels while recalibrating, to account for the drift in gyroscope data.

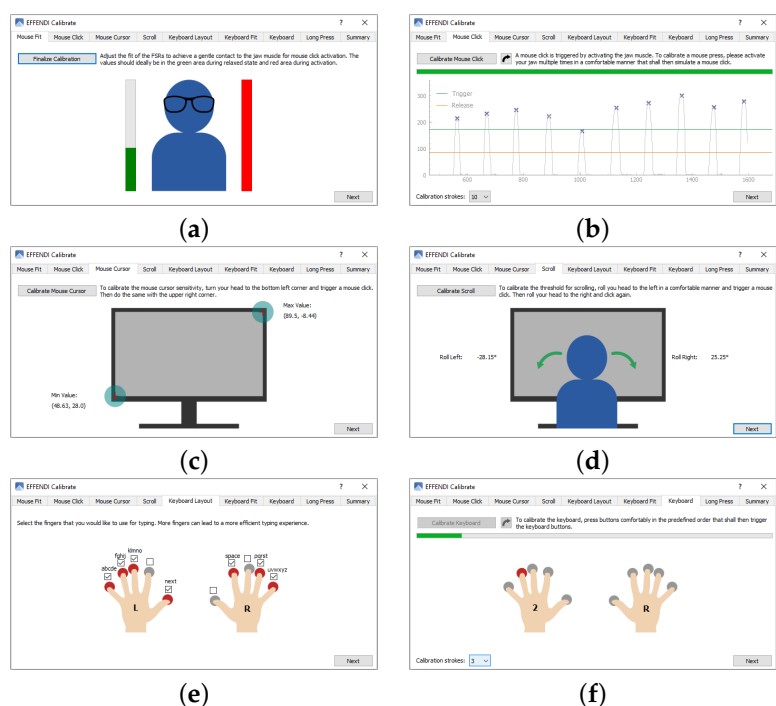

**Figure 8.** User interface during the guided calibration process. (**a**) Calibration of the force between the sensor and the Temporalis muscle. (**b**) Calibration of the threshold to activate a mouse click. (**c**) Calibration of the cursor sensitivity based on screen size. (**d**) Calibration of thresholds for scrolling. (**e**) Calibration of the keyboard layout based on usable fingers. (**f**) Calibration of the duration from a long button press.

Calibration of the scroll function works similarly to the user being asked to trigger mouse clicks with the alternative mouse while rolling the head first to the left and then to the right in a comfortable manner, as shown in Figure 8d. The determined values are then multiplied by a factor of 0.8 to serve as a threshold from here on.

Next, the user proceeds with the calibration of the alternative keyboard. To account for the variations in limb mobility, the user first selects the fingers that can be used to operate the alternative keyboard as in Figure 8e. This will then automatically determine letter and function distribution between the usable buttons, similar to Figure 6. It should be noted that the letter distribution will also affect the keyboard input code for each word during the generation of the dictionary. Once this step has been completed, the initial pressure that is exerted by the fingers in a resting state is measured to determine the new baseline signal, similar to the first step of the mouse calibration. The signals are collected after a 5-s resting phase.

The first tests showed that the force exerted by each individual finger can highly vary based on the preferred hand position. Thus, in the next calibration step, the trigger

threshold is individually determined for each keyboard button. The user is therefore asked to perform a button press 10 times with each finger in a comfortable manner by following the instructions on the screen, as seen in Figure 8f.

As a last calibration step, the user is asked to define a long button press, which is later used to trigger the secondary function of each button. The duration for a long press is determined by the average duration of the 10 long button presses the user was previously asked to execute. Finally, the user-specific calibration data can be saved in a file and be used for all future sessions with the alternative mouse and keyboard interface.

### 2.4. Desktop Application

A desktop application was developed to allow for PC usage that provides similar functionalities to a regular mouse and keyboard with common software. This was not possible with the previous implementation by Gür et al. [9], where the alternative keyboard could only be used in a confined testing environment.

When starting the application, a setup window appears on the screen, as shown in Figure 9, which now gives the user the option to either load previously saved personal parameters from the calibration process or adjust them manually. Here, the user also has the option to load a custom dictionary that is adapted to the individual text entry by favoring more frequently used words from among the word suggestions. The custom dictionary can also contain additional words that the user manually added during past usage. Accepting the parameters will trigger the IMU calibration and either generate a dictionary based on the chosen language or load an existing one.

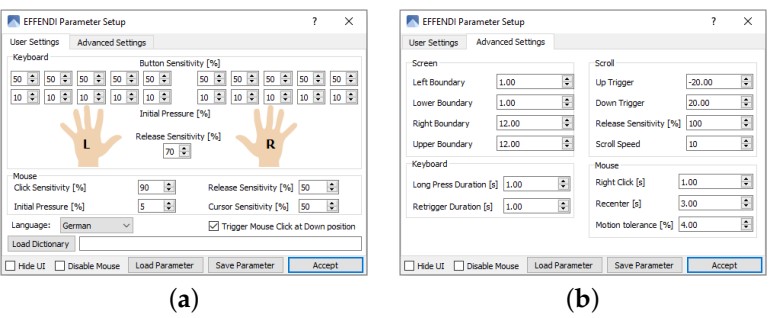

**Figure 9.** Setup window of the desktop application to load previously calibrated user data or perform manual changes. (**a**) User settings to adjust keyboard and mouse sensitivity. (**b**) Advanced settings to further customize user parameters.

The desktop application's user interface (UI) will then appear, which is always set to stay in the foreground above all other windows and has an opacity of 80 % to keep content behind the UI visible, which can be seen in Figure 2. The UI will appear at the bottom center, right above the taskbar, which is also shown in Figure 2. The UI now gives the user the option to make use of useful hotkeys, including *copy*, *paste*, *cut*, *select all*, *save*, *revert*, *find*, and *print*. In addition, the UI provides special characters and numbers that can be accessed through the alternative mouse, as shown on the left of Figure 10. Furthermore, the UI shows the suggested words in the center block, which are for the keyboard input of the user that is shown right above. In case a word is not contained in the dictionary, the user now has the option to add it manually through an on-screen keyboard. The user will have to click on the "+" symbol next to the field with the keyboard input. The UI will then be extended to the right with an on-screen keyboard, where words can now be added by typing the desired word using the alternative mouse, followed by a subsequent click on the save button in the lower-right corner. By using the button with the two arrows arranged in a circle in the lower-left corner of the on-screen keyboard, the user can save the updated word frequencies to the currently used dictionary.

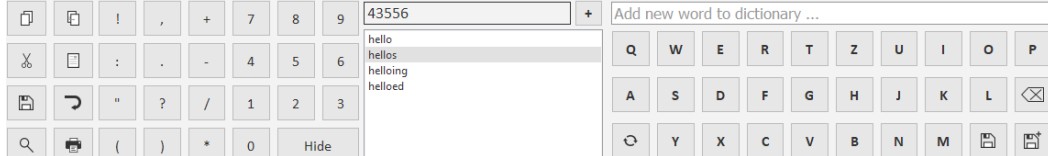

**Figure 10.** User interface of the desktop application with the extended on-screen keyboard on the right to add new words to the dictionary. Shortcuts are accessible through the first two columns, while the following six columns contain special characters and numbers. The middle section is used to display the entered keyboard input on top and to list the word suggestions in the white box below.

## 3. Study Design and Experimental Setup

To test the usability of the interface, three participants without disabilities, including one female and two males with an average age of $27 \pm 1$, were asked to execute tasks in a study using the regular mouse and keyboard as well as the alternative mouse and keyboard interface; they were also asked to compare the performance of each system. The experimental setup is shown in Figure 2. As a first step, the participants were asked to complete the previously described user-specific calibration process. This was followed by a click test with the alternative and regular mice as well as a typing test to compare the performance of the alternative and regular keyboards. Malheiros et al. [19] found that patients with Duchenne muscular dystrophy are able to increase their computer task performance through practice. Thus, the tests with the alternative mouse and keyboard were performed multiple times to potentially reveal a learning curve. Each of the tests was followed by a Raw-TLX (Task Load Index) [20] questionnaire to measure the user load. The Raw-TLX questionnaire is based on the original NASA-TLX [21] but omits the individual weighting of the subscales [20]. The subscales include the mental demand, physical demand, temporal demand, performance, effort, and frustration, which are rated on a scale of 100 points with 5-point increments [21]. Since the weighting of the subscales of the original NASA-TLX is still subject to scientific debate [22], Raw-TLX was chosen to measure the perceived workload of the proposed system and the given tasks.

The proposed interface was subsequently tested in real-world applications to examine its usability in everyday life. For repeatability across all participants, the users were introduced to each of the tasks through a short video. Finally, the participants were asked to complete a custom questionnaire that addressed their overall experience with the system and asked about potential improvements that could be made.

### 3.1. Mouse Test

To generate comparable results for the mouse click test, the tool FittsTaskTwo [23,24] was used to assess performance while considering Fitts' law [25]. Figure 11 shows the click test, with multiple circles arranged into a larger circle. The next circle to be clicked is filled in with color. A correct click results in a click sound, while an incorrect click results in a beep tone, providing direct feedback to the participant. One click test consists of four sessions, where the size *W* of each circle to be clicked varies between 30 and 90 pixels, and the size *A* of the circle they are arranged around varies between 500 and 750 pixels, as in [7]. It should be noted that the size and resolution of the screen impact the actual size of the objects on the screen, which is kept constant in this study. The click test is executed once with the regular mouse and three times with the alternative mouse to potentially reveal a learning curve.

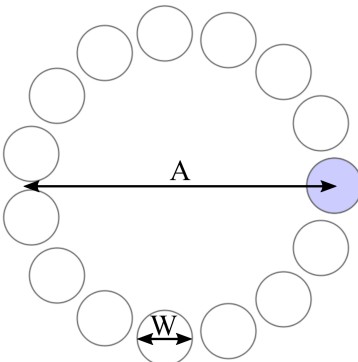

**Figure 11.** FittsTaskTwo [23,24] click test with the target's diameter W set to 90 pixels and the amplitude A set to 500 pixels in this example. The target to be clicked is the circle filled in with color.

### 3.2. Keyboard Test

The typing test was created using the website "10fastfingers.com" [26] and can be seen in Figure 12. The phrases used for the test were based on the ones from MacKenzie and Soukoreff [27], which were specifically designed for the evaluation of text entry techniques and were translated into German for this study. The phrases were divided into six text blocks, one for each session. The typing test was case-sensitive but did not include any punctuation marks or special characters. Participants were asked to execute the test once with the regular keyboard and five times with the alternative keyboard. Each session had a duration of two minutes. The order of the sessions was pseudo-randomized by shifting the order of the tests by one for each participant in order to counteract the potentially different levels of difficulty of the text blocks. For the evaluation, the number of correct and false words was counted. In addition, the number of entered words per minute (WPM) was determined by dividing the number of correctly entered characters by five in order to account for the varying word lengths in each of the phrases.

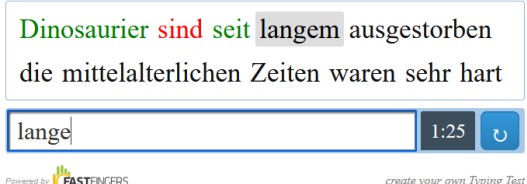

**Figure 12.** Typing test that was custom-created through the website "10fastfingers.com" [26], with the remaining time shown at the bottom right. Correctly entered words are marked in green and false words are marked in red.

### 3.3. Real-World Test

The real-world test was designed to investigate the performance of the alternative mouse and keyboard in daily life applications and included web browsing, working with a text editor, and making use of the onscreen hotkeys to speed up the workflow. The individual tasks can be summarized as follows:

- Open the browser and make the search query "hallo welt" (English: hello world);
- Open a Microsoft Word document on the desktop with a double click or a single click and the Enter key;
- Scroll to the end of the document by tilting the head to the right;
- Mark, copy, and paste the following sentence in the document using the on-screen keyboard shortcuts: "Willkommen am Lehrstuhl für Autonome Systeme und Mechatronik" (English: Welcome to the Chair of Autonomous Systems and Mechatronics);
- Type the following sentence, taking into account punctuation marks as well as upper and lower case letters: "Es gibt nichts Schöneres, als einen Schatz auf Svalbard zu

entdecken" (English: There is nothing better than discovering a treasure on Svalbard). In case one of the words is not in the dictionary, add it manually;
- Save the document through the shortcut on the on-screen keyboard.

During the test, the screen was recorded to enable a more accurate evaluation of the individual tasks with respect to time and accuracy.

### 3.4. User Confidence and Opinions

Beyond the Raw-TLX, a custom questionnaire was created at the end of the study to measure the user's experience of the system with regard to comfort, usability, and performance. To put the results into context, participants were asked about the ways and frequencies in which they used computers. Participants were also asked how they typically used the computer and how frequently to put the results into context. Moreover, open items allowed participants to express criticisms and suggestions for improvement of the interface.

### 4. Evaluation and Discussion

Three participants without disabilities participated in this study to investigate the performance of the alternative mouse and keyboard interface as well as the perceived workload compared to a regular computer mouse and text-entry system. Furthermore, the aim was to investigate the overall usability, especially focusing on daily life tasks. Information on anthropomorphic data such as the age, gender, or handedness of the participants is listed in Table **??**. All three participants reported using the computer several times a day, including the mouse and keyboard. The computer is mainly used for editing text documents, writing e-mails, and web browsing.

**Table 2.** Anthropomorphic data and calibration data of participants. Columns two to four contain the anthropomorphic data of each participant. Calibration data are provided in columns five to seven as M ± SD, with the values for mouse click and keyboard calibration being abstract nonlinear force values measured by the FSRs within a range of 0–1000.

|   | Age | Gender | Handedness | Mouse Click | Keyboard | Long Press (s) |
|---|-----|--------|------------|-------------|----------|----------------|
| 1 | 29 | male | right | $168.35 \pm 9.23$ | $516.07 \pm 38.15$ | $2.00 \pm 1.55$ |
| 2 | 27 | male | left | $128.07 \pm 10.88$ | $507.14 \pm 45.28$ | $0.51 \pm 0.06$ |
| 3 | 28 | female | right | $285.4 \pm 21.02$ | $516.06 \pm 57.08$ | $0.95 \pm 0.11$ |

### 4.1. Calibration

Calibration took 20 to 30 min for each participant and was crucial for the interface to work reliably. Based on the feedback from the custom questionnaire, the calibration process was rated to be intuitive and not too long. The mean (M) and standard deviation (SD) of the calibration data with 10 repetitions for mouse clicks, key presses, and long presses are listed in Table **??**. As described in the previous section, the calibration data of the mouse click and key press contain abstract force measurements, with FSR values between 0 and 1000, while the long press was measured in seconds. SD was approximately within 10% of the mean value of all measurements, except for the long presses for participant 1. Thus, 10 repetitions appear to be sufficient for a reliable calibration.

### 4.2. Mouse Test

All participants executed the click test once with the regular mouse and three times with the alternative mouse. The difficulty of each task within one session was determined by Shannon's formulation of the index of difficulty (*ID*), which is expressed as follows:

$$ID = log_2\left(\frac{A}{W} + 1\right), \tag{3}$$

and which depends on the target size *W* and Amplitude *A* shown in Figure 11 and is measured in bits [24]. Another relevant parameter with which to measure the performance

in the click tests is the throughput (*TP*) with Crossman's adjustment for accuracy [28], as expressed in the following equation:

$$TP = \frac{ID_e}{MT}. \tag{4}$$

The unit of *TP* is bits/s , *MT* is the time the participant needs to move the cursor between targets, and $ID_e$ is the effective index of difficulty [24], which equates to

$$ID_e = log_2\left(\frac{A_e}{W_e} + 1\right), \quad W_e = 4.133 \cdot SD_x, \tag{5}$$

with $W_e$ as the effective target width, $SD_x$ as the standard deviation of the pointer positions, also referred to as accuracy, and $A_e$ as the effective target amplitude along the task axis [29].

Setting the amplitude *A* to 500 and 750 pixels and the target width *W* to 30 and 90 pixels resulted in four different tasks with varying IDs within each session. Figure 13a shows the mean MT and error rate (ER) of all participants for different IDs. The movement time MT between the targets is much lower for the regular mouse compared to the alternative one, as expected for participants without disabilities. However, MT increased for both input devices for higher difficulties. The regular mouse had an error rate close to zero, and the alternative mouse had an error rate of up to 25%, with a rising trend for higher difficulties. In Figure 13b, the throughput and error rates are mapped over the three sessions performed with the alternative mouse. The single sessions performed with the regular mouse are displayed as lines across all sessions to serve as a reference. The regular mouse outperformed the alternative one with a higher TP and lower ER across all sessions. The alternative mouse performed reasonably well, with an average TP of 1.59 bits/s . It outperformed the IMU headset by Fall et al. [7] and could thus be a good alternative for people with motor disabilities. However, no increase in performance can be seen for any input device across the sessions.

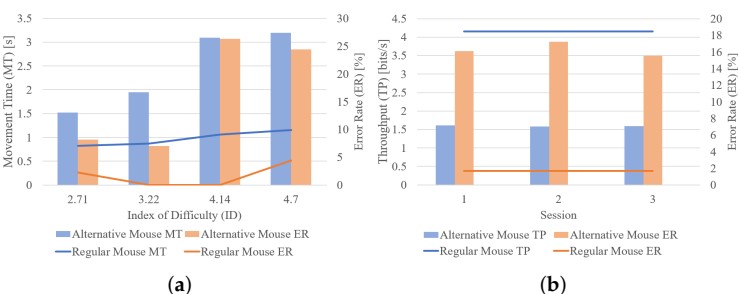

**Figure 13.** Graphs to compare the performance of the alternative mouse to the regular one. (**a**) MT and ER plotted over ID for the alternative and regular mice. (**b**) TP and ER plotted over sessions for both mouse input devices.

### 4.3. Keyboard Test

Figure 14 shows the average amount of WPM and the ER plotted over the five sessions for each input device. ER is derived by dividing the falsely entered words into the sum of correctly and falsely entered words. For the first two sessions, the ER was much higher when the alternative keyboard was used as compared to the regular one. Since the alternative keyboard was new to all participants, this was to be expected as users first need to adapt to the system. For sessions three to five, this changed in favor of the alternative keyboard, which had a lower ER as compared to the reference value of the regular keyboard. It should be noted that typical typos did not lead to an increased ER for the alternative keyboard due to the user selecting the desired word from a compiled suggestion list. A typical typo would thus lead to more time needed per entered word, resulting in less WPM, while the ER would only be affected by miss-selections from the

word suggestion list. Overall, the performance of the alternative keyboard, with an average of 8.44 WPM over sessions three to five, is much lower compared to the reference value of 37.67 WPM, but it outperforms the scanning ambiguous keyboard of MacKenzie and Felzer [13], reaching 5.11 WPM for the last text block. Moreover, it is promising to see an improvement by a factor of 2.30 with respect to WPM for the alternative input device when comparing the average performance of sessions three to five to the first session. With more practice, it is expected that even better performance will be seen once the user familiarizes themselves with the system by memorizing the letters assigned to each key, thus allowing them to focus solely on the word suggestion list.

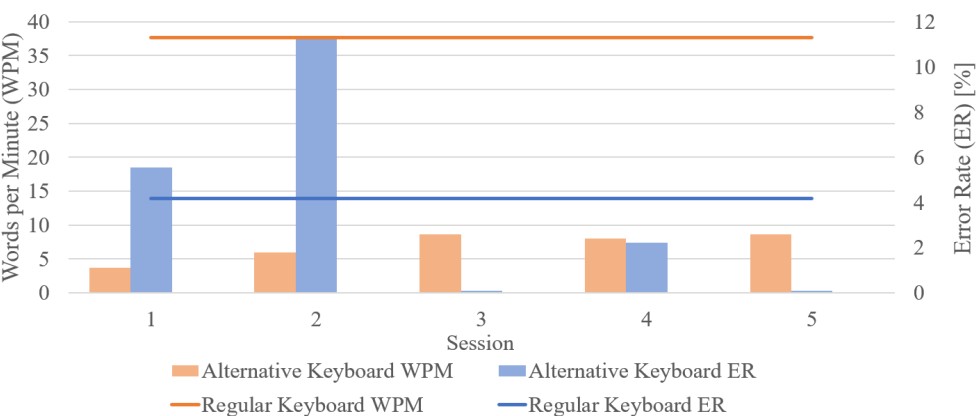

**Figure 14.** WPM and ER plotted for sessions using the alternative and regular keyboards.

The dictionary created for each participant constantly adapts by favoring more frequently used words in the suggestion list. It is unlikely that this had an impact on the performance during the limited time of the experiment. Through longer use, however, the constant adaptability of the system could increase the WPM by allowing the user to make fewer keystrokes until the desired word appears under the word suggestions, with the most frequently used words always appearing on top.

### 4.4. Overall System

The results of the six Raw-TLX subscales gathered after each experiment are shown in Figure 15. The perceived workload was rated significantly higher across all subscales for the alternative input devices compared to the standard versions. The total perceived workload was computed by averaging all subscales and resulted in values of 36.39 and 41.11 for the alternative mouse and keyboard, respectively, which is considered to be "somewhat high" according to the interpretation score in [30]. Furthermore, the participants experienced high effort, high mental demand, high temporal demand, and somewhat high physical demand with the alternative input devices. A decrease in mental demand might be expected over time due to familiarization with the system. All three participants criticized the pointer precision of the alternative mouse. Frustration was, however, rated relatively low compared to the other subscales and shows the overall reliability of the proposed interface. The system calibration seems to strongly impact the reliability of the system to a point where poor calibration can lead to an unusable interface for a user. A second calibration iteration after gaining first experiences with the alternative input device could increase predictability and thus increase user experience. It is likely that temporal demand was partially caused by the time constraint of the experiments and would probably be lower during normal usage. Finally, the performance achieved with the alternative system was perceived to be relatively good by the participants compared to the other subscales.

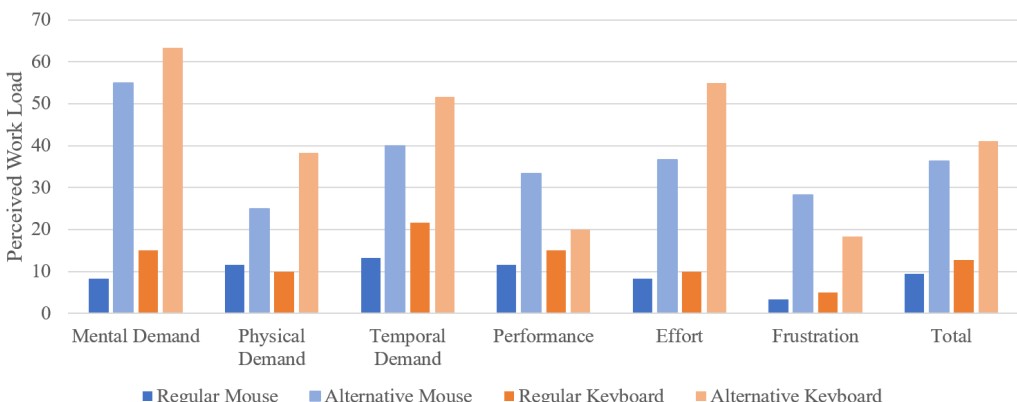

**Figure 15.** Perceived workload for six subscales of the Raw-TLX questionnaire, for the alternative and regular input devices.

The real-world test showed that the proposed interface allows the user to perform all the given tasks and is thus applicable for daily life. This is in line with the statement by Participants 1 and 3 in the custom questionnaire, that the system would be usable in daily applications with the current functionalities; Participant 2 chose a neutral position on that statement. The user interface of the desktop application, shown in Figure 10, was reported to be intuitive by the first two participants, with Participant 3 having a neutral opinion. Furthermore, Participant 2 suggested an increase in the size of the shortcut buttons of the user interface.

Overall, the participants liked the fit of the spectacle frame and were not distracted by triggering mouse clicks through the activation of the masticatory muscle. However, Participant 3 criticized the inaccuracy of the mouse clicks, which had been caused by involuntary movements of the IMU through the activation of the Temporalis muscle, and suggested that the IMU be detached from the spectacles to overcome this issue. There are multiple possibilities to resolve this unwanted behavior. The IMU could, for instance, be attached to the user's head with something similar to a headband, which could remove those artifacts. However, according to the case study of Abrams et al. [8], this might not find acceptance from the target group. A quick solution to overcome this challenge could be the triggering of the mouse click upon contraction of the Temporalis muscle instead of relaxation, as it is currently implemented. This is expected to reduce the impact of the involuntary movement of the IMU on the mouse click accuracy. First tests with this setup already showed promising results and could be investigated in a following study. Another promising long-term solution could be the contact-free detection of the Temporalis muscle contraction by proximity sensors. This approach was already tested for dietary monitoring and eating event detection by Saphala et al. [31,32] and Selamat et al. [33] and could be adapted to our interface with little effort.

Finally, the positioning of the keys and the key input, including haptic feedback, were also rated as neutral to comfortable on average. Participant 3 would like to have the ability to erase full words with the alternative keyboard and to be able to make corrections on individual letters within a word.

The results shown in Figures 13 and 14 indicate that working with the alternative interface is slower than with a regular mouse and keyboard, which aligns with the subjective questionnaire feedback. All participants would not choose the alternative system over the regular one. This was to be expected from users without disabilities since the alternative interface was designed for people with limited upper limb mobility and was intended to support people who find it difficult to operate a regular mouse and keyboard for more efficient use of the computer. To match the capabilities of the target group, certain design choices were made, which might be counterproductive for people without disabilities; for example, the change from a tactile keyboard with a single function per key to a keyboard with multiple functions per key with FSR-based buttons. Additionally, the participants

were most likely experienced in using the regular mouse and keyboard. With additional training on the alternative system, the performance gap could potentially be narrowed. However, the results confirm the basic functionality of the alternative interface and its potential use for the target population.

## 5. Conclusions

This work presents a highly personalizable alternative mouse and keyboard interface for people with motor disabilities of neuromuscular origin. Our approach focused on the adjustability of the interface through hardware and software to account for differing symptomatologies of the target population. The alternative mouse in the form of spectacles adapted to various head shapes and sizes, which made the device more accessible than previous implementations. The use of a higher-quality IMU improved the reliability of cursor control through head motion tracking, while the use of smaller FSRs reduced the bulk of the device, making it more comfortable to wear.

The mechanical button design of the alternative keyboard was improved to achieve better finger positioning while typing and to assure the good visibility of the naming indicators at all times. By placing the keys on the magnetic pads, the user could easily reposition the buttons for more comfortable use. Furthermore, the addition of multiple language settings increased the number of potential users, while favoring the more frequently used words by the user could lead to a more efficient typing experience.

The added hardware adaptability is backed up by a software calibration pipeline that adjusts sensor sensitivity and considers the individual needs of each user. The use of FSRs in general, for both mouse click activation and the detection of key presses, allows the calibration of the trigger threshold to especially account for the needs of individuals who have lost fine motor control. Most importantly, the desktop application makes the interface easy to use in daily life, allowing the user to perform tasks such as text editing or web browsing.

The general functionality of the interface was tested in a small study with three participants without disabilities while focusing on the requirements of the target group. The perceived workload was rated as "somewhat high" by the participants, which is promising considering the limited time they spent familiarizing themselves with the proposed interface. However, it should be noted that the perceived workload of the alternative interface was rated to be much higher by the participants, with the largest differences having been observed in mental demand and effort. This could possibly be explained by the fact that the participants were not familiar with the alternative interface yet. We expect a decrease in mental demand and effort over longer periods of usage. Although the performance of the alternative mouse and keyboard interface was much lower during the click and typing tests compared to the regular mouse and keyboard, this needs to be evaluated for the target group.

In future studies, we plan to include more people without disabilities to test the functionality of new features, and up to five people from the target group to investigate the effectiveness of the system. To assure good comparability, we intend for people from the target group to perform the same tests as the three participants without disabilities that took part in this study. The identified demands of the target group by Abrams et al. [8] were considered during the design process of our proposed alternative interface. We expect similar performance with the alternative mouse and keyboard, while the performance with the regular interface will likely be lower for people with limited upper limb mobility. Through its novel degree of personalization, the proposed alternative mouse and keyboard interface shall provide increased accessibility to various user groups, almost independent of their anatomy and physiology.

**Author Contributions:** Conceptualization, D.A. and P.B.; methodology, D.A., H.S., A.B. and P.B.; software, D.A.; validation, D.A., H.S., A.B. and P.B.; formal analysis, D.A., H.S. and A.B.; investigation, D.A. and H.S.; resources, D.A. and P.B.; data curation, D.A., H.S. and A.B.; writing—original draft preparation, D.A.; writing—review and editing, D.A., A.B. and P.B.; visualization, D.A.; supervision,

P.B.; project administration, D.A. and P.B.; funding acquisition, P.B. All authors have read and agreed to the published version of the manuscript.

**Funding:** This research work was funded by the German Research Foundation (DFG), grant number FE 936/6.

**Institutional Review Board Statement:** The study was conducted according to the guidelines of the Declaration of Helsinki and in accordance with recommendations from the Institutional Ethics Committee of Friedrich-Alexander-Universität Erlangen-Nürnberg (21-495_1-S 2022-02-07).

**Informed Consent Statement:** Written informed consent was obtained from all subjects involved in the study.

**Data Availability Statement:** The data presented in this study are available on request from the corresponding author. The data are not publicly available due to privacy and ethical reasons.

**Acknowledgments:** The authors would like to thank all participants who took part in this study.

**Conflicts of Interest:** The authors declare no conflicts of interest.

## Abbreviations

The following abbreviations are used in this manuscript:

| | |
|---|---|
| FSR | Force-sensitive resistor |
| IMU | Inertial measurement unit |
| UI | User interface |
| TLX | Task load index |
| WPM | Words per minute |
| M | Mean |
| SD | Standard deviation |
| ID | Index of difficulty |
| TP | Throughput |
| ER | Error rate |

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
