# Peer review of "Design and Implementation of a Personalizable Alternative Mouse and Keyboard Interface for Individuals with Limited Upper Limb Mobility"

_mti, doi:10.3390/mti6120104_

Round 1

Reviewer 1 Report

The authors have proposed the human interface device combining functionality of keyboard and mouse dedicated to users with impaired mobility of upper limb.
The work of the author is significantly based on the work by D. Gür and others (doi:10.3390/mti4040084) cited as [9].
The contribution of the authors is oriented on adaptation of the solution for the particular user.
The adaptability is implemented in hardware (modifiable layout of ambiguous keyboard sensors, flexible positioning of the temporalis muscle sensor) and software (individual calibration, configurable set of ambiguous keyboard keys, configurable dictionary). Additionally the authors have modernized the hardware selecting newer components offering better sensitivity and accuracy.
It seems that the improvement provided by the authors is sufficient to justify publication as a new research paper.
Comparing to the work [9] the authors have performed more detailed assessment of performance of their solution.
However, the obtained results may be biased by the fact that the device was tested only by users without disabilities.
Especially the information about the additional effort and demand related to use of the alternative mouse/keyboard obtained from the able-bodied users may be misleading.
Testing the device by users with disabilities could provide much better results, and I'd strongly suggest performing such tests an publishing their results (the author propose it for "further studies", but at least preliminary results could be included in that paper).

The text requires some language corrections. Below are just a few from abstract:
line 1 "what" -> ", which"
line 4 "whith a neuromuscular..."
line 5-6 "personalization on hardware and software level" -> "personalization on the hardware and software levels"
line 7 "frame with integrated motion sensor" -> "frame with an integrated motion sensor"

Author Response

Thank you very much for the helpful comments. The response letter is atteched.

Reviewer 2 Report

The manuscript presents a mouse/keyboard device designed preferably for users with limited upper limb mobility. A functionality test is presented. However, members of the target group were not included in this test, which gives limited evidence as to whether this design is suitable for the target group. 

Design and evaluation are well performed. However, I suggest improving the description of the device. This description should first describe the overall functionality, before mentioning implementation details, such as processors and other components. At a first glance, it was not clear to me what belonged to the device and what belonged to the test setup. Further, it should be highlighted what the novelty of this design consists of; it seems that it is a combination of the design of the mouse alternative, as well as the arbitrary position of keys on a magnetic plate, in connection with spelling software.

There is a variety of complex sentences that hinder the reading flow. E.g., Line 31: "... which avoids them ...": is "avoids" the correct word here?

Line 80: please introduce the abbreviations at their first occurrence. (It was very helpful to find a list of abbreviations at the end of the manuscript!)

Line 88: ... redesigned ...: Please provide a description of what has been redesigned, as the following text refers to many details that have been altered from the previous design. This description appears somewhat unstructured. Further, it remains unclear whether some of these design decisions are more or less important than others.

Line 431ff: these sentences are difficult to understand. On several occasions, it is difficult to grasp what belongs to what. E.g., The 25% seem to refer to the alternative mouse, but probably the error rate is meant here. Also, Line 436 is too complex.

Figure 14 (also 13): The graph(s) show the values for the alternative implementation as bars while the values for the regular implementation are shown as a line. As long as this is a threshold, this is ok. However, if these lines represent measured values, it does not make sense to have the session number on the x-axis (as this is not a continuous space).

Line 511: is "migrated" the correct word here? Maybe: adapted?

Line 514ff: complicated sentence. "... stated to prefer to have ... to erase ... " is difficult to understand.

Line 517ff: complicated sentence. Suggestion: "The results shown in Figures 13 and 14 indicate that working ..."

Line 520: Is there any reasoning for why this could be expected?

The conclusion should be better informed by test results and how to interpret these. Further, it must be mentioned how a test with persons of the target group would be performed, and what to expect from such an evaluation. This additional research will give proof that your design is suitable for the target group.

Author Response

(The authors gave the same response as above.)
